# Detection of Tomato Leaf Miner Using Deep Neural Network

**DOI:** 10.3390/s22249959

**Published:** 2022-12-17

**Authors:** Seongho Jeong, Seongkyun Jeong, Jaehwan Bong

**Affiliations:** Department of Human Intelligence Robot Engineering, Sangmyung University, Cheonan-si 31066, Republic of Korea

**Keywords:** artificial neural network, deep neural network, plant disease detection, tomato leaf miner, classification, segmentation, real-world agricultural site

## Abstract

As a result of climate change and global warming, plant diseases and pests are drawing attention because they are dispersing more quickly than ever before. The tomato leaf miner destroys the growth structure of the tomato, resulting in 80 to 100 percent tomato loss. Despite extensive efforts to prevent its spread, the tomato leaf miner can be found on most continents. To protect tomatoes from the tomato leaf miner, inspections must be performed on a regular basis throughout the tomato life cycle. To find a better deep neural network (DNN) approach for detecting tomato leaf miner, we investigated two DNN models for classification and segmentation. The same RGB images of tomato leaves captured from real-world agricultural sites were used to train the two DNN models. Precision, recall, and F1-score were used to compare the performance of two DNN models. In terms of diagnosing the tomato leaf miner, the DNN model for segmentation outperformed the DNN model for classification, with higher precision, recall, and F1-score values. Furthermore, there were no false negative cases in the prediction of the DNN model for segmentation, indicating that it is adequate for detecting plant diseases and pests.

## 1. Introduction

Protection of crops from plant disease is a problem that is intertwined with agriculture and climate change [1]. Climate change caused by global warming alters host resistance and pathogenic rates, as well as affects physiological interaction between hosts and pathogens [2]. The plant disease problem has worsened as various plant diseases spread faster than ever before around the world. The possibility of plant diseases emerging in previously non-affected regions has increased, and there is a lack of local expertise to treat the new plant disease in the regions [3].

Tomato leaf miner causes crop losses of around 80 to 100% in places where it is present. By invading leaves, stems, flowers, and fruit, the tomato leaf miner destroys the structure of the tomato growth. It is extremely difficult to prevent the spread of the tomato leaf miner. Despite significant efforts to prevent its migration, the tomato leaf miner took only about three years to spread across southern Europe after it was first identified in Spain. Now the tomato leaf miner is found in most South American countries, Southern Europe, Northern Africa, and West Asia [4].

Leaves, the most vulnerable component of a plant, are where disease symptoms first appear [5]. From the very beginning of their life cycle until they are ready to be harvested, the crops need to be inspected in a timely manner to protect the crops against various plant diseases. Agricultural specialists conventionally observed agricultural fields using the time-consuming approach of naked-eye surveillance to keep a check on the plant leaves for symptoms of diseases [6].

In agriculture, computer vision tools have largely supplanted the naked eye to identify plant diseases and pests. Computer vision tools frequently employed conventional image processing algorithms, which need human feature design along with classifiers to detect plant diseases and pests. Computer vision tools increase detection performance by creating imaging schemes and selecting appropriate light sources and shooting angles based on the characteristics of plant diseases and pests.

Although handcrafted imaging schemes help computer vision tools detect plant diseases and pests, they also increase the application cost. Furthermore, in a complex natural environment, plant diseases and pests are difficult to identify via handcrafted computer vision tools because it is hard to expect that the traditional computer vision tools completely exclude the influence of low contrast, large variations in the scale, image noise, and disturbances under natural light [7].

The capacity to directly use raw data without using a handcrafted feature extractor is a significant advantage of deep neural network (DNN) models [8]. DNN models especially based on convolutional neural networks (CNN) have shown success in recent years when used in a variety of computer vision applications, such as traffic detection, medical image recognition, scenario text detection, and face recognition [9,10,11,12].

Several DNN-based approaches for detecting plant diseases and pests have been studied using leaf images. The DNN-based approaches can be further separated into a classification method, a detection method, and a segmentation method according to the types of output [13]. The classification method produces the types of plant diseases and pests [14,15,16,17]. The detection method provides the location, as well as the types of plant diseases and pests [18,19,20,21,22,23,24,25,26,27,28,29,30,31,32]. The segmentation method yields the types of plant diseases and pests, as well as pixel information, such as location and geometric properties [33,34,35,36].

In this study, DNN-based approaches for the classification and the segmentation methods were used to diagnose the tomato leaf miner. Two DNN models were trained using the same RGB images of tomato leaves captured from real-world agricultural sites. The diagnosis performance of two DNN models was evaluated and compared. One of the two DNN models was suggested as for tomato leaf miner detection based on the diagnosis performance.

## 2. Materials and Methods

### 2.1. Dataset Description

AI Hub is operated by National Information Society Agency in the Republic of Korea to accelerate the advancement of artificial intelligence technology and its application. Various data were released on AI Hub related to natural language, healthcare, autonomous driving, agriculture, livestock, education, and so forth.

The Agricultural Knowledge Base (AKB) dataset [37], one of the agricultural datasets released on AI Hub, was organized by I IMC corporation in 2018. The AKB dataset contains a total of 40,704 RGB image data of rose leaves and tomato leaves taken in the laboratory and real-world agricultural sites. All leaf images of the AKB are labeled with normal and types of diseases. The rose and tomato leaves in the AKB dataset are labeled with 11 and 17 types of plant diseases, including normal leaves.

We processed the AKB dataset in two different ways to train and evaluate two types of deep neural networks (DNNs) applicable to real-world agricultural sites. Images and labels of normal and mined tomato leaves collected from real-world agricultural sites were selected from the AKB dataset. In the selected dataset, there are 3115 and 3341 pairs of images and labels of normal tomato leaves and mined tomato leaves, respectively. Images in the selected dataset had various sizes. All the images in the selected dataset were resized to 300 by 300 pixels. The selected and resized dataset (*D_RN152_*) was used to train and evaluate a DNN model for image classification. The *D_RN152_* was separated into a training dataset, a validation dataset, and a test dataset while the split ratio was 60%, 20%, and 20%, respectively.

The region of tomato leaf infected by the tomato leaf miner was segmented into a polygonal shape by the human from the resized images in the AKB dataset to generate binary mask images. The binary mask images have a size of 300 by 300 pixels, which is the same as that of the resized images. Pixel values where the tomato leaf miner occurred in an image were converted into one. The other pixel values in the image were converted into zero. When all the pixel values in the image were converted into one or zero using the previously described method, the image was changed to the binary mask image.

Pairs of the resized image and the binary mask image (*D_MRCNN_*) were used to train and evaluate a DNN model for object segmentation. The *D_MRCNN_* was separated into a training dataset, a validation dataset, and a test dataset while the split ratio was 60%, 20%, and 20%, respectively.

### 2.2. Deep Neural Network for Tomato Leaf Miner Classification

Transfer learning is one type of machine learning method. Transfer learning makes use of previously learned knowledge from a different problem that is applicable to a new problem. Additionally, the knowledge is applied to solve the new problem.

ResNet [38] was developed using a residual learning framework with a shortcut structure to address the issue of DNN performance degrading as depth exceeds a certain number of layers. ResNet152, one of the ResNet structures, was pre-trained on the ImageNet dataset, which contains over 14 million images and 1000 different labels.

It was assumed that the classification process of *D_RN152_* is relevant to that of the ImageNet dataset. ResNet152 was used for transfer learning to classify the *D_RN152_* as a binary class of the normal tomato leaf and the tomato leaf infected by the tomato leaf miner. Figure 1 shows the developed DNN structure for processing *D_RN152_* using transfer learning with ResNet152 (*DNN_RN152_*).

Structure of *DNN_RN152_* is shown in Figure 1. In Figure 1, green, blue, grey, and orange boxes denote the pooling layer, convolutional layer, residual module, and fully connected layer, respectively. Red lines indicate shortcut structures for residual learning. *DNN_RN152_* has two individual stages to process the images: feature extractor and classifier. From *conv 1* to *conv 5* in Figure 1 are the feature extractors that extract feature maps from the images. The fully connected layers (FCLs) in Figure 1 are the classifier that makes a prediction using the feature maps.

Layers of the *DNN_RN152_* from *conv 1* to *conv 5* reused the structure and weights of the ResNet152 trained on the ImageNet dataset. The structure of the FCLs was determined by finding optimal hyperparameters using Bayesian optimization (BO). BO is one of strategy for finding a set of hyperparameters from a hyperparameter space to optimize an objective function that requires a large amount of computational power and, thus, is expensive to evaluate. Table 1 shows a hyperparameter space explored to determine the structure of the FCLs and training process. The objective function of the BO was set to an F-1 score of the validation dataset, which indicates the performance of classification (see Equation (3) for more detail). The hyperparameter space was iteratively explored by the BO method to find the optimal set of hyperparameters that maximize the F-1 score.

As a result of the BO, the number of layers, the number of neurons and dropout rate in each layer were determined for the FCLs as 1.64 and 0.2, respectively. The rectified linear unit and the softmax were used for the activation function in the hidden layer and the output layer. The number of neurons in the output layer was one to deal with the binary classification. For the learning process, the batch size, the optimizer, and the learning rate were set to 64, SGD, and 0.001, respectively.

The *DNN_RN152_* was trained twice using the training dataset of the *D_RN152_*. For the first training process, the weights of the feature extractor were frozen and not trained. Only the weights of the classifier were trained. All the weights of the *DNN_RN152_*, both the feature extractor and the classifier, were trained during the second training process.

### 2.3. Deep Neural Network for Tomato Leaf Miner Segmentation

A DNN model for segmentation (*DNN_MRCNN_*) was trained in a transfer learning manner using Mask R-CNN [39], a type of region-based convolutional neural network. The Mask R-CNN was pretrained on the COCO dataset and implemented using Matterport’s library [40] in the TensorFlow environment. The *DNN_MRCNN_* was developed to segment and classify regions infected by tomato leaf miner from a leaf image by training the Mask R-CNN using *D_MRCNN_*.

The *DNN_MRCNN_* processed a leaf image as shown in Figure 2 and yielded class, confidence, bounding box, and binary mask features for segmented pixels.

The feature maps of the input leaf image were extracted using ResNet101, denoted by (a) in Figure 2. In Figure 2b, a region proposal network [41] generates anchor boxes in regions expected to contain plant diseases. In Figure 2c, the predicted anchor boxes and the feature maps were processed to generate fixed-size feature maps by using the region of interest pooling method [42]. The output of the region of interest pooling method was used as input for two types of DNNs—FCL and feature pyramid networks (FPN) [43]. The FCL, in Figure 2d, classified the anchor boxes as the tomato leaf miner or the background and predicted the position of bounding boxes for the tomato leaf miner. The FPN, in Figure 2e, generated a binary mask image with a value of 1 for the regions infected by the tomato leaf miner, and a value of 0 for the remaining regions.

The *DNN_MRCNN_* was trained using the *D_MRCNN_*. The batch size, optimizer, and learning rate were set to 2, SGD, and 0.001, respectively, during the learning process. Loss and MRCNN class loss on the training and validation dataset are shown in Figure 3. During the learning process of Mask R-CNN, five types of losses are computed. A MRCNN class loss is one of the five losses and represents how successfully the Mask R-CNN classifies the detected object from the image. A loss refers to the aggregate of all five losses. The loss and MRCNN class loss on the training dataset are shown in Figure 3a,b, respectively. Figure 3c,d represent the loss and MRCNN class loss on the validation dataset. Even after 25 epochs, the loss and MRCNN class loss on the training dataset tend to decrease. Whereas the loss and MRCNN class loss on validation dataset do not improve after 5 epochs, but rather deteriorate. The Mask R-CNN was trained until 5 epochs, where the loss and MRCNN class loss on the validation dataset had the lowest value.

When there were regions with a value of 1 in the binary mask image predicted in the input leaf image, it means that tomato leaf miner-infected regions were detected in the input leaf image. In this case, the input leaf image was classified as a tomato leaf miner infection. The input leaf image was classified as normal in the opposite case.

### 2.4. Performance Evaluation Metrics for Developed Deep Neural Networks

The performance of two developed DNNs for the tomato leaf miner was evaluated using four metrics of confusion matrix, precision, recall, and F1-score. The DNN for the tomato leaf miner segmentation was additionally evaluated using intersection over union (IoU).

In the classification problem, the confusion matrix compares the prediction results of DNN with the target value and presents the comparison results in matrix form. There are four types of comparison results: true positive (TP), false positive (FP), false negative (FN), and true negative (TN). When the regions infected by the tomato leaf miner are actually present in the leaf image, the TP is the case where the DNN predicts that the leaf image contains the infected regions. The FN is the case in which the DNN predicts that there is no infected region in the leaf image. The FP is the case where the DNN predicts the leaf image contains the infected regions when all of the leaves in the leaf image are normal and do not contain the infected region. The TN is the case in which the DNN predicts that there is no infected region in the leaf image. In other words, the TP and TN are the cases in which the DNN prediction and the target value match. The FP and FN are the cases when those are not matched.

The precision is the rate at which the leaf image actually contains the infected regions when the DNN predicts the leaf image as the infection of tomato leaf miner. The precision was calculated by Equation (1) using the number of the TP and the FP cases denoted by *n(TP)* and *n(FP)*, respectively.
(1)$$Precision=\frac{n\left(TP\right)}{n\left(TP\right)+n\left(FP\right)}$$

The recall is the probability that the DNN prediction and the target value are matched when the leaf image actually contains the infected regions. The recall was calculated by Equation (2) using the number of the TP and the FN cases denoted by *n(TP)* and *n(FN)*, respectively.
(2)$$Recall=\frac{n\left(TP\right)}{n\left(TP\right)+n\left(FN\right)}$$

The precision and the recall have an inverse relationship. The F1-score is the harmonic mean of the precision and the recall, and it is used to reflect both the precision and the recall in the DNN performance evaluation for classification. The F1-score was calculated by Equation (3) using the calculation results of the precision and the recall denoted by *cal(Precision)* and *cal(Recall)*, respectively.
(3)$$F1-score=2*\frac{cal\left(Precision\right)*cal\left(Recall\right)}{cal\left(Precision\right)+cal\left(Recall\right)}$$

The IoU represents the percentage of matches between the human-segmented binary mask image and the DNN-predicted binary mask image. To calculate the IoU in the plant leaf images, an overlapping area and a union area between the plant disease regions segmented by the human and the plant disease regions predicted by the DNN are required. The human-segmented and DNN-predicted plant disease regions were polygonal, making it difficult to calculate their numerical area. The number of pixels in the overlapping area and the union area between the human-segmented and DNN-predicted plant disease regions were counted to substitute the area calculation.

Both the human-generated and the DNN-generated binary mask images were converted into two-dimensional matrices consisting of integers 0 and 1 to be used in counting the number of pixels in the overlapping and union areas. The sum of the two matrices was used to count the pixels in the union area between the human-segmented plant disease regions and the DNN-predicted plant disease regions. Elements with values one and two in the sum result of the two matrices were counted as the number of pixels in the union area. Then, the Hadamard product was computed between the two matrices to count the pixels in the overlapping area between the human-segmented plant disease regions and the DNN-predicted plant disease regions. Elements with the value one were counted as the number of pixels in the overlapping area as a result of the Hadamard product. The IoU was calculated by Equation (4) using the counted number of pixels in the overlapping area and the union area. In Equation (4), the number of pixels in the overlapping area and the union area are denoted by *n(A_O_)* and *n(A_U_)*, respectively.
(4)$$IoU=\frac{n\left(A_{O}\right)}{n\left(A_{U}\right)}$$

## 3. Results

The performance of the *DNN_RN152_* and the *DNN_MRCNN_* was evaluated using the test dataset. On both the *DNN_RN152_* and the *DNN_MRCNN_*, the confusion matrix, the precision, the recall, and the F1-score were calculated. The IoU, on the other hand, was calculated only for *DNN_MRCNN_*.

### 3.1. Confusion Matrix

The *DNN_RN152_* and the *DNN_MRCNN_* were both evaluated using the test dataset, which included 623 normal leaf images and 665 leaf images with infected regions of the tomato leaf miner. The *DNN_RN152_* directly classified the input tomato leaf images as normal or infected with tomato leaf miner. The *DNN_MRCNN_*, on the other hand, segmented the regions infected by the tomato leaf miner, and the segmentation results were used to classify the input tomato leaf images, as explained in Section 2.3.

Two confusion matrices in Figure 4a,b describe the classification results using *DNN_RN152_* and *DNN_MRCNN_*, respectively. As shown in Figure 4a, the *DNN_RN152_* classified the 665 leaf images with infected regions of tomato leaf miner into 560 infected leaf images and 105 normal leaf images. The 623 normal leaf images were classified into 89 normal leaf images and 534 infected leaf images using the *DNN_RN152_*. In Figure 4b, the *DNN_MRCNN_* correctly classified all 665 leaf images with infected regions of tomato leaf miner as infected leaf images. The 623 normal leaf images were classified into the 594 normal leaf images and the 29 infected leaf images using the *DNN_MRCNN_*. Using the *DNN_RN152_*, 649 of 1288 images, or 51.7%, were classified to match the true and DNN-predicted classes. Using the *DNN_MRCNN_*, 1259 of 1288 images, or 97.7% of the total, were classified as matching between the true and DNN-predicted classes.

The *DNN_MRCNN_* predicted fewer cases of FN and FP than the *DNN_RN152_*. It is important to note that *DNN_MRCNN_* does not predict FN, which suggests that *DNN_MRCNN_* is more adequate than *DNN_RN152_* in actual application for plant disease and pest detection.

### 3.2. Precision, Recall, and F1-Score

The precision, recall, and F1-Score were calculated based on the results of the confusion matrix. Figure 5 compares the precision, recall, and F1-Score calculated from the confusion matrices of the *DNN_RN152_* and the *DNN_MRCNN_*. In Figure 5, red and blue bars denote the performance evaluation results of the *DNN_RN152_* and the *DNN_MRCNN_*, respectively. The precision, the recall, and the F1-Score calculated from the prediction results of the *DNN_MRCNN_* all showed higher values than those calculated from the prediction results of the *DNN_RN152_*, indicating that the *DNN_MRCNN_* outperforms the *DNN_RN152_* in terms of diagnostic performance of the tomato leaf miner.

### 3.3. Intersection over Union

IoU was calculated by comparing the human-segmented binary mask image and the DNN-predicted binary mask image. Figure 6 shows the calculation results of IoU. The human-segmented binary mask image and the DNN-predicted binary mask image are superimposed in Figure 6b. Yellow denotes the human-segmented plant disease region. In addition, the DNN-predicted plant disease region is depicted in yellow with high translucency. The overlapping images are arranged based on the IoU value.

A bar graph in Figure 6a shows the number of test datasets with IoU values within a certain range. The horizontal axis in the bar graph represents the range of IoU values with a minimum value of 0 and a maximum value of 1 divided by 0.1 intervals. The vertical axis indicates the number of *DNN_MRCNN_* predictions whose IoU value corresponds to the range of IoU values on the horizontal axis. The minimum and maximum IoU values for *DNN_MRCNN_* prediction results on test datasets were 0.05 and 1.0, respectively. In addition, the average of the IoU values was 0.59, with a variance of 0.03.

When the IoU value is 0.6 or higher in Figure 6b, two human-segmented and DNN-predicted binary mask images are nearly identical. As shown in Figure 6a, approximately half of the prediction results for 665 images containing the leaf infected by tomato leaf miner have an IoU value of 0.6 or higher. Except for 50 predictions, the vast majority have an IoU of 0.4 or higher. The IoU calculation results suggest that *DNN_MRCNN_* is capable of precisely locating lesions.

## 4. Conclusions

In this study, we developed two DNN models to diagnose the tomato leaf infected by the tomato leaf miner. *DNN_RN152_*, one of the developed DNN models, employed the well known convolutional neural network structure, ResNet152. *DNN_RN152_* made use of a feature extractor same to the ResNet152 and a customized classifier. Using the tomato leaf image as input, *DNN_RN152_* directly classified the image as the normal leaf or the leaf infected by the tomato leaf miner. The Mask RCNN was used to develop another DNN model, *DNN_MRCNN_*. The regions infected by tomato leaf miner were segmented from the tomato leaf image using *DNN_MRCNN_*, and the segmented results were used to classify the tomato leaf image as the normal leaf or the leaf infected by tomato leaf miner.

The same tomato leaf images captured from real-world agricultural sites were used to train and evaluate both DNN models. The human-segmented binary mask images were additionally provided to the *DNN_MRCNN_* for the training process.

As a preliminary study, we compared the performance of DNN models for classification (*DNN_RN152_*) and segmentation (*DNN_MRCNN_*) to determine which DNN model is better for detecting single plant disease in a single crop from images captured in real-world agricultural sites. The precision, recall, and F1-score were used to assess the performance of the two developed DNN models. For all criteria, the *DNN_MRCNN_* outperforms the *DNN_RN152_* in terms of the diagnostic performance of the tomato leaf miner. The IoU was additionally calculated to assess the segmentation performance of the *DNN_MRCNN_*. The IoU calculation results showed that in the majority of test datasets, and the *DNN_MRCNN_* precisely segmented the regions infected by tomato leaf miner from the input image.

In future work, we intend to train the DNN model using Mask RCNN to detect multiple plant diseases and pests that occur in various crops in real-world agricultural sites.

## Figures and Tables

**Figure 1 sensors-22-09959-f001:**
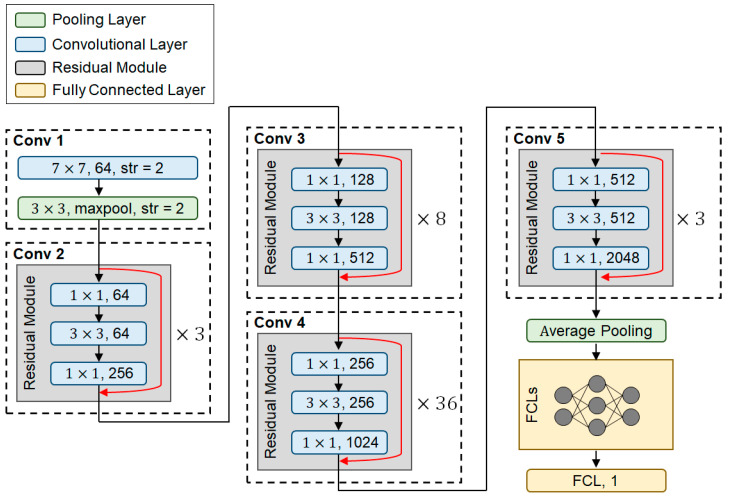
Developed DNN structure using transfer learning with ResNet152.

**Figure 2 sensors-22-09959-f002:**
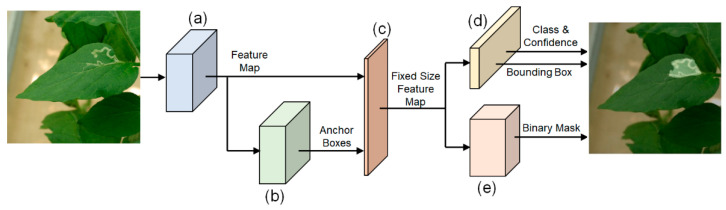
Developed DNN structure using transfer learning with Mask R-CNN. (**a**) ResNet101, (**b**) Region proposal network, (**c**) Region of interest pooling, (**d**) Fully connected layer, (**e**) Feature pyramid networks.

**Figure 3 sensors-22-09959-f003:**
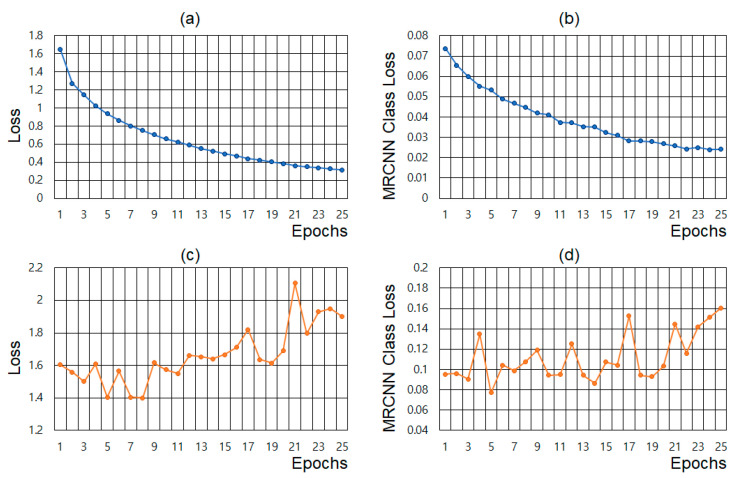
Learning curves for Mask R-CNN: (**a**) training loss, (**b**) training mrcnn class loss, (**c**) validation loss, and (**d**) validation mrcnn class loss.

**Figure 4 sensors-22-09959-f004:**
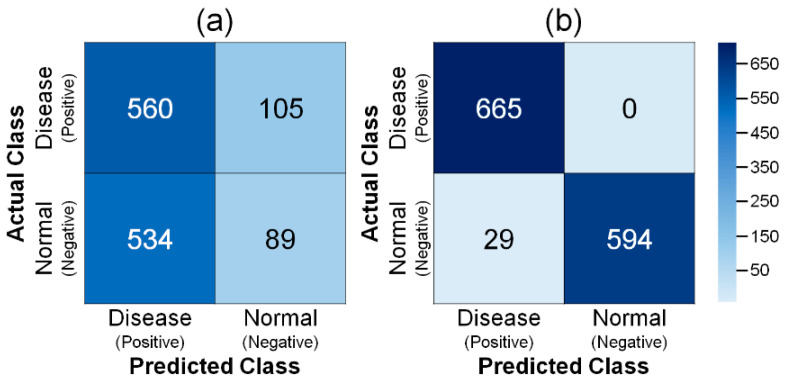
Confusion matrices of two DNNs: (**a**) *DNN_RN152_*, (**b**) *DNN_MRCNN_*.

**Figure 5 sensors-22-09959-f005:**
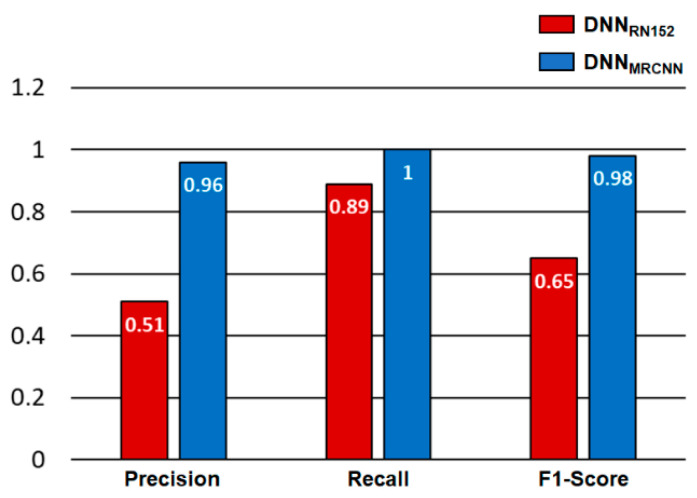
Comparison of performance evaluation results between two DNNs—*DNN_RN152_* (red bar) and *DNN_MRCNN_* (blue bar).

**Figure 6 sensors-22-09959-f006:**
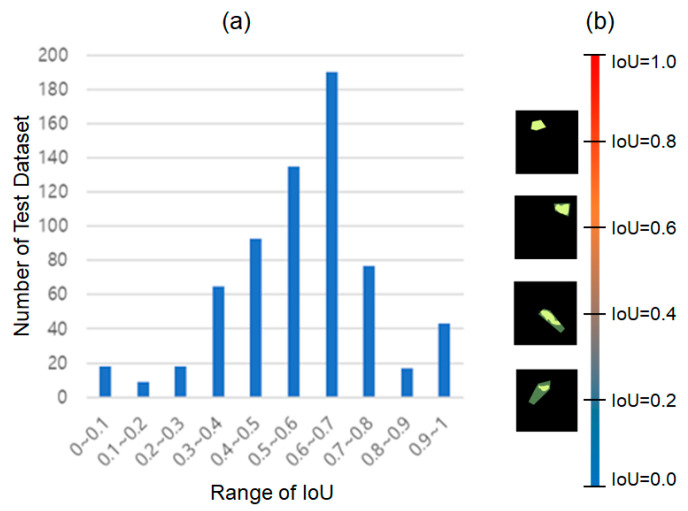
Number of test datasets (vertical axis) corresponding to the IoU range (horizontal axis) calculated by comparing human-segmented and DNN-predicted binary mask images for test datasets. (**a**) Number of test dataset corresponding to the IoU range calculated from prediction, (**b**) Comparison between target masking and predicted masking according to IoU value.

**Table 1 sensors-22-09959-t001:** Explored hyperparameter space.

Types of Hyperparameter		Explored Range
FCL ^1^ Structure for Classifier	Number of Layers	Integer in a range of [0, 4]
	Number of Neurons in Each Layer	2^n^, where *n* is an integer in a range of [4, 8]
	Dropout Rate in Each Layer	Real number in a range of [0.1, 0.5] with an interval of 0.1
Training Process	Batch Size	2^n^, where *n* is 3 or 4
	Optimizer	One among SGD, Adam, and RMSprop

^1^ Fully Connected Layer.

## Data Availability

Not applicable.

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
