# Peer review of "Detection of Tomato Leaf Miner Using Deep Neural Network"

_sensors, 2022, doi:10.3390/s22249959_

Round 1

Reviewer 1 Report

The authors presented DNN techniques based on DNN segmentation and classification of potato leaf miners. The authors showed a comprehensive framework of the design of the ML model and the choice of performance metrics to evaluate the performance of the model. The overall idea of the article sounds well and helps in the field of smart agriculture. Hence, some minor comments could be addressed to improve the quality of the paper:

1) Figures need to be smaller to make the article looks prettier. Especially figure 6 please use a high-quality confusion matrix 

2) Could you please elaborate more on the choice of the CNN structure and how you managed to ensure no overfitting or bias due to the imbalanced dataset?

3) Some figures showing the Epochs, validation, training accuracy, recall, and F_1 score could be added to visually examine the robustness of the CNN-trained model.

I would like to thank the authors for their effort on overall effort in employing AI techniques for smart agriculture which will help in the future adoption of the automated farming 5.0 paradigm shift.

I suggest addressing the comments before the final acceptance of the article for publication.

Best regards

Reviewer 2 Report

The author have presented a method for plant disease detection. The method is developed for tomatto leaf miner detection. The method is based on a neural network. 

Figure-8 is not clear. It is unclear what is on the x and y axis. There are two type of DNN models. Transformer based and CNN based. The author need to mention why CNN based model has been chosen as appose to tansformer based model. Why the authors chose a size of 300 x 300. and not 500x500. Please explain human labeling of leaf miner disease. There is not reason to show a binary bask of a healthy leaf. They author need to explain why the binary mask is not accuracte only to cover leaf disease. They have also covered the health part of the leaf too. The diagram needs to be corrected in Figure-3. A standard classification architecture has three blocks, Stem block, feature block, and classification head. Why the classification head used in Figure-3 is using 1000 classes model? Is not there is only 1 class? Why did you use ResNet152 for this simple 2 class classification task? Why not use a simple resNet-18? I suppose a simple ResNet-18 will work here. If not then we should compare both and discuss that in the discussion session. Figure-4 is the actual model you have used, then what is the purpose of Figure-3? It is redundant and confusing the readers. Also have you tried using the initial weights from the ImageNet? why not. What is the purpose of an MLP here when a simple FCC can work? The ground truth images are not clear. You need to show more images for clarification. 

The author used only 1 disease for this experiment. The model comparison is not very clear. The novelty is also not clear. A comparison of two techniques on a public dataset for one single disease is itself does not carry much weight in terms of the contribution to the body of the knowledge.   

Author Response

Because the answer was lengthy, I attached it as a word file.

Round 2

Reviewer 2 Report

The authors have improved significantly their manuscript. However, it still requires further language improvement. 

Author Response

The authors did our best to improve the quality of English throughout the paper.